# Twist-angle engineering of excitonic quantum interference and optical nonlinearities in stacked 2D semiconductors

Kai-Qiang Lin[1✉], Paulo E. Faria Junior[1], Jonas M. Bauer[1], Bo Peng [2], Bartomeu Monserrat [2,3], Martin Gmitra[4], Jaroslav Fabian[1], Sebastian Bange [1] & John M. Lupton [1✉]

Twist-engineering of the electronic structure in van-der-Waals layered materials relies predominantly on band hybridization between layers. Band-edge states in transition-metal-dichalcogenide semiconductors are localized around the metal atoms at the center of the three-atom layer and are therefore not particularly susceptible to twisting. Here, we report that high-lying excitons in bilayer WSe$_2$ can be tuned over 235 meV by twisting, with a twist-angle susceptibility of 8.1 meV/°, an order of magnitude larger than that of the band-edge A-exciton. This tunability arises because the electronic states associated with upper conduction bands delocalize into the chalcogenide atoms. The effect gives control over excitonic quantum interference, revealed in selective activation and deactivation of electromagnetically induced transparency (EIT) in second-harmonic generation. Such a degree of freedom does not exist in conventional dilute atomic-gas systems, where EIT was originally established, and allows us to shape the frequency dependence, i.e., the dispersion, of the optical nonlinearity.

[1] Department of Physics, University of Regensburg, Regensburg, Germany. [2] TCM Group, Cavendish Laboratory, University of Cambridge, Cambridge, UK. [3] Department of Materials Science and Metallurgy, University of Cambridge, Cambridge, UK. [4] Department of Theoretical Physics and Astrophysics, Pavol Jozef Šafárik University, Košice, Slovakia. ✉email: kaiqiang.lin@ur.de; john.lupton@ur.de

Excitons are objects of volume. When they arise in two-dimensional monolayer semiconductors, transition and binding energies become particularly sensitive to the immediate surrounding[1]. Whereas dielectric effects are straightforward to rationalize, the impact of artificial van-der-Waals stacking[2] of two monolayer crystals is dominated by the underlying electronic orbitals: electronic bands hybridize[3–6], and moiré bands[7–10] and moiré excitons[11–15] form. To first order, the optoelectronic properties of transition-metal dichalcogenide (TMDC) monolayers[16] are dominated by the electronic transition between the highest-energy valence bands and the lowest conduction bands (CBs) at the K-points of the Brillouin zone. Such dipole-allowed transitions form strongly bound band-edge A-excitons (X)[1]. Under conditions of strong optical driving, higher-lying electronic bands need to be considered in the optoelectronic response of monolayer $WSe_2$[17–19]. In essence, electrons can be promoted to an upper CB at the K-points, i.e., the lower spin-split CB+2 band, while staying Coulombically bound to the hole: high-lying excitons (HXs) form[20]. With respect to the ground state, the excitonic equivalent of a ladder-type three-level atomic system is then established which exhibits quantum interference when driven by pulsed laser radiation[17]. Such HX species coincidently appear around twice the X energy[20] so that quantum interference can occur between X and X → HX transitions, leading to electromagnetically induced transparency (EIT) in an optical second-harmonic generation (SHG)[17].

Here, we demonstrate that HX is ten times more susceptible to band hybridization than X, shifting by over 230 meV with twist angle in bilayer $WSe_2$. This tuning allows twist-angle engineering of EIT, which enables us to turn the quantum interference in twisted-bilayer $WSe_2$ on and off. We demonstrate the universality of this concept by showing how EIT, which occurs in monolayer $WSe_2$ but not in $MoSe_2$, arises in bilayers of the latter under appropriate twisting. Such tunability, which goes beyond conventional atomic-gas systems[21,22], strongly modifies the frequency dependence of the second-order susceptibility and offers a route to engineering optical nonlinearities.

## Results

**Twist-angle control and characterization.** We tune the transition energies of the low-energy X and high-energy HX excitons by depositing two $WSe_2$ monolayers on top of each other, twisted by an angle $\theta$ as sketched in Fig. 1a. The transition energy of X can be easily identified in monolayer $WSe_2$ and twisted bilayers by low-temperature photoluminescence (PL) under non-resonant excitation as illustrated in Fig. 1b. To suppress electronic transitions from regions in reciprocal space other than the K-points, high-energy HX PL can be generated by an Auger-type upconversion process[19,20] through pumping resonantly at the X transition as indicated in Fig. 1c. We define the twist angle of the bilayer deterministically by successive dry-transfer[23] of two segments of the same single-crystal monolayer onto a Si/SiO2 substrate. A representative bilayer of $WSe_2$ fabricated with $\theta = 45°$ is shown in Supplementary Fig. 1. As explained in Supplementary Figs. 1 and 2, this angle can be measured on non-overlapping parts of the individual monolayers after sample fabrication by exploiting the strong dependence of the intensity of the co-polarized SHG component on the offset between laser polarization direction and crystallographic orientation[4,12,14]. Note that interlayer sliding, strain gradients, and atomic reconstruction may be introduced during the stamping process, which would be expected to give rise to inhomogeneous broadening with increasing imaging spot size.

**Twist-angle dependence of the band-edge A-exciton X.** We begin by probing the energy of X in artificial $WSe_2$ bilayers at a

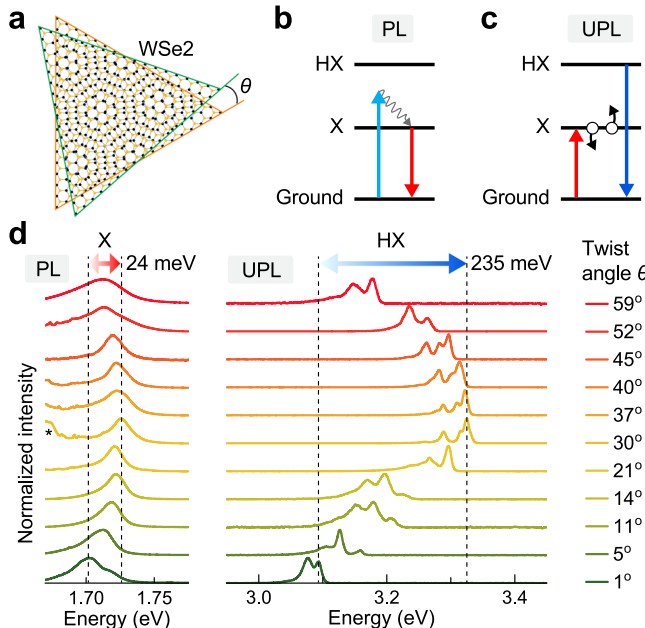

**Fig. 1 Twist-angle dependence of the A-exciton (X) and the high-lying exciton (HX) photoluminescence (PL) at 5 K. a** Illustration of artificial twisted-bilayer $WSe_2$ with a stacking angle of $\theta$. **b** Origin of the A-exciton PL (red arrow) under 488 nm continuous-wave excitation (blue arrow). The gray arrow indicates nonradiative relaxation. **c** Origin of the upconverted HX PL (UPL) (dark blue arrow) arising through an Auger-like exciton–exciton annihilation process (black circles and arrows) under resonant pumping of the A-exciton (red arrow). **d** Variation of the A-exciton PL (left panel) and HX UPL (right panel) with twist angle $\theta$. The energy of X only varies within a 24 meV range as marked by the double-headed red arrow and dashed lines, while the energy of HX varies over 235 meV as marked by the blue arrow. The intervalley exciton is marked by an asterisk.

temperature of 5 K, with stacking angles varying between 0° (3R stacking) and 60° (2H stacking). In monolayer $WSe_2$, X is associated with the direct bandgap at the K-points and gives rise to strong PL emission. When the layer number increases, this excitonic transition shifts slightly to the red, and intervalley indirect-bandgap excitonic transitions appear[24]. Figure 1d shows the PL of the excitonic ground state X:1s in twisted-bilayer $WSe_2$ under excitation by the 488 nm line of an argon-ion laser. An artificial $\theta = 59°$ bilayer, which is almost identical to natural 2H-$WSe_2$ bilayers, shows emission peaking at 1.71 eV. Decreasing the twist angle to 30° increases the interlayer distance and shifts X to the blue[25,26], i.e., towards the energy of the A-exciton PL of single-layer $WSe_2$[20]. For the present analysis, we ignore signatures of intervalley excitons[6,27,28] (marked by an asterisk in Fig. 1d). Further decrease of the twist angle to 1°, shifts the energy back towards the red. Note that, due to the three-fold crystal symmetry, the $\theta \approx 60°$ bilayer is not equivalent to the $\theta \approx 0°$ bilayer, the X PL of which is shifted further to the red. This asymmetry stems from the suppressed interlayer hybridization in 60° (2H) stacking due to the antiparallel spin ordering of the spin-split bands in the two layers. As a consequence, the interlayer hole-hopping is blocked in 60° stacking because of the large spin–orbit coupling in the valence band[6].

**Twist-angle dependence of the high-lying exciton HX.** Next, we probe the energy of the HX by pumping with a tunable continuous-wave laser set to a photon energy of 1.72 eV, in resonance with the X:1s transition. Following the population of X, electrons are excited to a high-lying conduction band through

an Auger-like exciton–exciton annihilation process. Figure 1d shows the upconverted PL (UPL) spectrum from the HX[20], which consists of a zero-phonon peak and a phonon progression. The spectrum may also possibly contain contributions from charged excitons. We identify the zero-phonon peak as the $s$-like HX state[20]. As for the case of monolayers[20], the power dependence of the HX UPL in bilayers is sub-parabolic because of bleaching of the A-exciton transition at high fluences, which constrains the yield of upconversion arising from the Auger-like exciton–exciton annihilation mechanism[19]. As shown in Supplementary Fig. 3, such an HX state in bilayer WSe₂ is also observable in one-photon, i.e., conventional Stokes-type, PL under illumination by a UV laser. In this case, however, the spectral sub-structure tends to be obscured by a broad background signal, which inevitably arises both from substrate luminescence and from optical transitions in the sample from regions in momentum space other than the K-points[20]. In contrast, pumping the A-exciton transition in UPL selectively excites the HX around the K-points and gives rise to a background-free HX PL spectrum[20]. The HX in UPL persists to temperatures above 125 K as shown in Supplementary Fig. 4. As for the X:1$s$ PL, the HX transition shifts to the blue between $\theta = 59°$ and $\theta = 30°$, and back to the red between $\theta = 30°$ and $\theta = 1°$. The scale of the twist-angle-dependent shift between $\theta = 1°$ and $\theta = 30°$ is ten times larger—235 meV for the HX transition compared to 24 meV for X:1$s$. This corresponds to an average twist-angle susceptibility of 8.1 meV/°, which is the largest value reported to date (see Supplementary Table 1 for a comparison). Such a large value cannot be explained merely by changes in Coulombic correlations with twisting[27], but matches well with the picture provided by the partial charge densities of electron states at the K-points. As shown in Fig. 2a, b, the electronic states of the high-lying conduction bands of monolayer WSe₂ do not localize close to the tungsten atom, in stark contrast to the case for the first conduction bands. The HX, which arises mainly from the lower spin-split CB+2[20], is therefore much more prone to hybridization between layers. Although it is challenging to resolve HX transitions in twisted bilayers directly from DFT calculations due to the appearance of many folded minibands, the dependence of HX on the twist angle resembles that of the variation of interlayer distance as shown in Supplementary Fig. 5. However, the HX of 1° stacking has a substantially lower energy than that of 59° stacking. This discrepancy cannot be rationalized by the interlayer distances. The asymmetry between $\theta \approx 60°$ and $\theta \approx 0°$ is equivalent to the situation for the X:1$s$ PL, but the scale of the asymmetry is larger for the HX. This increase may be a result of suppressed interlayer electron hopping for 60° (2H) stacking because of the large spin–orbit coupling in the CB+2 band. The strong twist-angle dependence of the HX PL, therefore, offers an additional method to assess twisting, complementary to the conventional technique based on SHG. It raises the accuracy particularly for small twist angles and distinguishes the 0° and 60° orientations.

**Twist-angle engineering of excitonic quantum interference.** The nonlinear-optical response is particularly sensitive to contributions from excitonic resonances[29], which can give rise to strong effects in the time and frequency domain, particularly under strong pump intensities where multiple Rabi cycles can occur[30]. But the situation is much subtler in WSe₂. With the two excitonic species X and HX sharing the hole state in the same valence band, optical excitation of electrons from the lowest conduction band to the higher-lying conduction band allows the dark $p$-like HX state to be populated under appropriate optical pumping[20]. Such a non-emissive $p$-like HX state was identified by two-photon excitation spectroscopy in monolayer WSe₂ at energy

slightly above the emissive $s$-like HX state[20]. The ground state, the X:1$s$ state, and the $p$-like HX state form the excitonic equivalent of an atomic multilevel system with states $|1\rangle$, $|2\rangle$, $|3\rangle$ as indicated in Fig. 2c. In this system, transitions $|1\rangle \rightarrow |2\rangle$ and $|2\rangle \rightarrow |3\rangle$ are dipole-allowed and can be driven by the same laser pulse centered at a frequency $\omega$, provided that the detuning from each of the resonances is sufficiently small. The $|1\rangle \rightarrow |3\rangle$ transition is forbidden. Under such conditions, quantum interference occurs between the two excitation pathways $|1\rangle \rightarrow |2\rangle$ and $|1\rangle \rightarrow |2\rangle \rightarrow |3\rangle \rightarrow |2\rangle$. This quantum interference counteracts the concurrent excitonic resonant enhancement and leads to a pronounced dip in the SHG spectrum of the WSe₂ monolayer[17]. Such an appearance of quantum-interference effects in the SHG is similar to the situation encountered in EIT[21,22], where a strong control beam induces transparency for an independent weak probe beam. Here, however, control and probe are degenerate, and quantum interference occurs for laser frequencies close to $\hbar\omega = (E_{12} + E_{23})/2$, where $E_{12}$ and $E_{23}$ are the energies for the $|1\rangle \rightarrow |2\rangle$ and $|2\rangle \rightarrow |3\rangle$ transitions, respectively.

To examine the effect of shifting energy levels on EIT in SHG, we compute the effect of quantum interference on the SHG using the time-dependent density-matrix formalism[17]. Figure 2d shows that the condition of quantum interference, manifested by an EIT dip in the calculated SHG spectrum, can be both shifted in energy and suppressed completely by detuning state $|3\rangle$. The energetic position of the spectral dip in the SHG closely aligns to the energy $E_{13}$ of state $|3\rangle$. As we demonstrate in the following, such a detuning can be achieved by twisting of bilayers.

Experimentally, a change of the dielectric environment also shifts energy levels, but this effect is, in general, not of sufficient strength to modify the underlying condition for quantum interference: $E_{12}$ and $E_{13}$ tend to shift together by the same amount[17,31]. Assuming that the energy of the $p$-like HX follows the $s$-like (i.e., radiative) HX as a function of twist angle $\theta$, seen in Fig. 1, the twist-related shifts of state $|3\rangle$ should be much larger than those of $|2\rangle$. Twisting would then give direct control of quantum interference. Figure 2e shows false-color plots of the normalized intensity of SHG radiation from bilayer WSe₂ as a function of the emitted photon energy and the mean pump-laser energy $\hbar\omega$ for a representative choice of twist angles. Analogous results for further angles are shown in Supplementary Fig. 6. The spectral dip caused by the quantum interference is clearly visible in the normalized SHG spectra and can indeed be turned on and off. In the absence of quantum interference, for both the 1° and 59° twist angles, the SHG spectrum follows the canonical linear dependence on excitation energy 2$\hbar\omega$. The anti-crossing feature associated with quantum interference[17] is recovered for twist angles between 21° and 45°. The EIT dip in the SHG spectrum of twisted-bilayer WSe₂ is generally shifted to the red by about 55 meV in comparison with that of bare monolayer WSe₂[17]. However, the depth of the dip cannot be interpreted directly as the coherence time of state $|3\rangle$ because local disorder may be introduced to different degrees during the stamping process. Nevertheless, the anti-crossing feature persists up to 125 K, close to the 150 K threshold found for monolayer WSe₂ (Supplementary Fig. 7). These temperatures are significantly higher than the 75 K threshold we reported for monolayers previously[17], which is likely a result of improved crystal quality. The temperature threshold of the quantum-interference feature can therefore be employed as a criterion to characterize the quality of both monolayers and stacked bilayers.

**Twist-angle engineering of optical nonlinear dispersion.** The absolute SHG intensities (which are proportional to the square of the second-order susceptibility) of the $\theta > 0°$ samples are

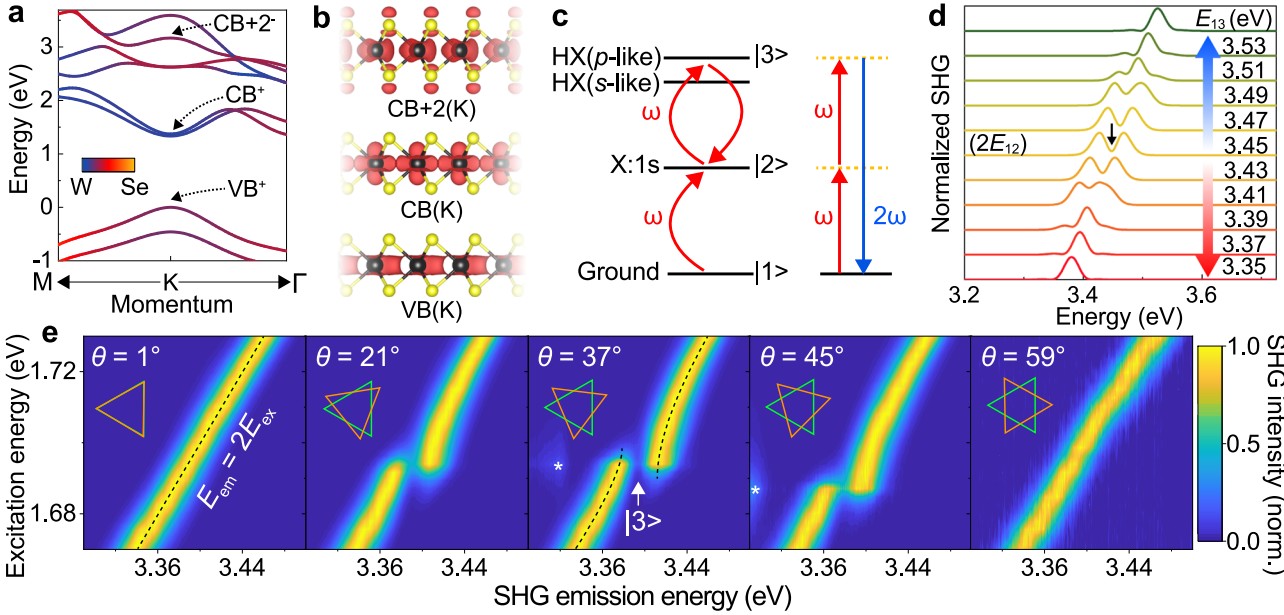

**Fig. 2 Twist-angle dependence of quantum interference in an optical second-harmonic generation (SHG) from bilayer WSe$_2$. a** Band structure of monolayer WSe$_2$ from DFT calculations. The X arises from transitions between the top valence band VB$^+$ band to the upper spin-split first conduction band CB$^+$, and the HX results mainly from transitions between VB$^+$ and CB+2$^-$. The color code indicates the wavefunction projection around each atom, illustrating the atomic-orbital contributions to the energy bands. At the K-points, the high-lying bands CB+2 have a much greater contribution from selenium atoms than the CB band. **b** Partial charge densities (red) of the states from VB, CB, and CB+2 at the K-points, which was calculated for monolayer WSe$_2$ without spin–orbit coupling. For states from the VB and CB, the charge is localized around the tungsten atoms (black spheres). For the state from CB +2, the charge appears to be more delocalized and extends to the selenium atoms (yellow spheres). In bilayers, the HX is therefore expected to be more sensitive to twisting than X when hybridization occurs between layers. The isosurface levels are set to 0.01 e/Bohr$^3$. **c** Illustration of the excitonic three-level system, where $|1\rangle$ is the ground state, $|2\rangle$ is associated with the 1s state of the A-exciton X:1s, and $|3\rangle$ is associated with the $p$-like high-lying exciton HX. Strong driving by an ultrashort laser pulse (red arrows) centered at a frequency $\omega$ dresses state $|2\rangle$ and drives quantum interference between the three states. At the same time, the laser pulse gives rise to SHG (blue arrow), which probes the interference. **d** As captured by time-dependent density-matrix simulations, tuning the energy of state $|3\rangle$ while keeping state $|2\rangle$ at 1.725 eV allows the quantum-interference feature, which shows up as a dip in the SHG spectrum, to be effectively turned on and off. The dip in the SHG spectrum marks the energy of state $|3\rangle$ as illustrated by the black arrow for the case of $E_{13} = 3.45$ eV. **e** Experimental twist-angle ($\theta$) dependence of normalized SHG from bilayer WSe$_2$ at 5 K, measured with 80 fs excitation pulses. The normalized SHG intensity is plotted as a function of emitted photon energy (horizontal axis, $E_{em}$) and central photon energy of the excitation laser (vertical axis, $E_{ex}$). The UPL feature is marked by asterisks. The SHG follows the canonical linear dependence on the excitation energy ($E_{em} = 2E_{ex}$, black dashed line in the left panel), as shown in the 1° and 59° twisted bilayers. Anti-crossing behavior (black dashed lines in the middle panel) becomes apparent when quantum interference occurs, as shown in the 21°, 37°, and 45° twisted bilayers.

generally expected to drop monotonically with increasing twist angle due to the associated increase of effective inversion symmetry[32]. This expectation is no longer valid when the excitonic resonances shift substantially and quantum interference occurs, as shown in Fig. 3. For instance, the 21° twisted-bilayer WSe$_2$ has a much larger second-order susceptibility than the 1° sample around 1.73 eV excitation energy.

The spectrum of the second-order susceptibility dispersion changes completely with twist angle, and the modulation in effective amplitude can reach close to two orders of magnitude. Resonant enhancement of the second-order susceptibility by excitonic transitions is usually limited to a specific range of frequencies. The ability to tune resonances allows the frequency dependence of the second-order susceptibility to be chosen to match a particular desired frequency range. In Fig. 4, we summarize the dependence on twist angle of the peak energy of the X:1s PL (i.e., state $|2\rangle$, red), the zero-phonon line energy of the $s$-like HX PL[20] (blue), and the energy of the EIT dip in the SHG spectrum (i.e., the $p$-like HX state $|3\rangle$, yellow). As for WSe$_2$ monolayers[17], in twisted bilayers, the energy of state $|3\rangle$ is slightly smaller than twice the energy of state $|2\rangle$. Quantum interference only appears when the detuning between the excitation laser frequency and both $E_{12}$ as well as $E_{23}$ is not too large. Shifting the $p$-like HX closer to an energy of $2 \times E_{12}$ at twist angles 21°–45°

enables quantum interference to occur in the bilayer. The maximum energy offset $\Delta E = E_{12} - E_{23} = 2E_{12} - E_{13}$ for which quantum interference is detectable is 68 meV here. This upper limit is somewhat higher than the value of 40 meV estimated from the density-matrix calculations of the simplified three-level model discussed in Fig. 2 using the experimental laser-pulse width of 23 meV, although this agreement should be considered reasonable given the model simplifications[17] compared to a full description using semiconductor Bloch equations. Bare monolayer WSe$_2$ and hBN-encapsulated WSe$_2$ have an experimental $\Delta E$ of 46 and 21 meV, respectively[17], which is well below the value of 68 meV established here with the twisted bilayer.

**Activation of excitonic quantum interference in MoSe$_2$.** Given the similarities in the band structure, excitonic multilevel systems capable of supporting EIT-type phenomena are expected to be a generic feature of TMDC materials such as MoSe$_2$, MoS$_2$, WS$_2$, and MoTe$_2$. However, the condition of a small $\Delta E$ necessary to enable quantum interference in SHG is not always fulfilled in native monolayers. For instance, monolayer MoSe$_2$ does not show quantum interference in SHG as seen in Fig. 5a, where the SHG emission spectrum energy follows the canonical linear dependence on pump photon energy. We propose that this absence of

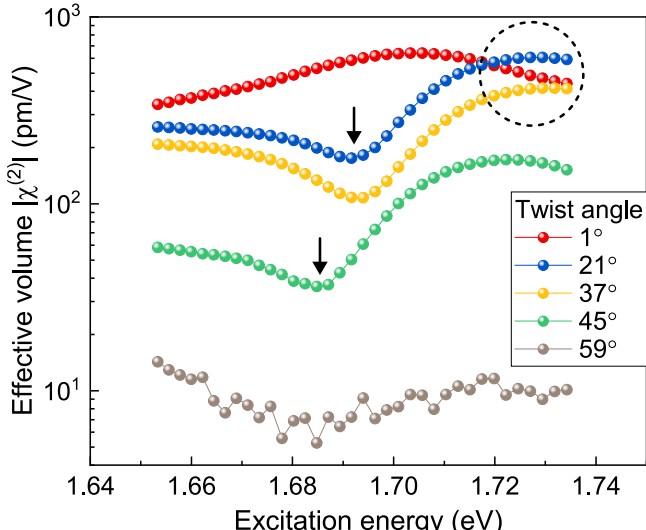

**Fig. 3 Experimental frequency dependence of the effective second-order nonlinear susceptibility as a function of twist angle in bilayer WSe₂ at 5 K.** The dashed circle highlights the region where the 21° twisted bilayer exceeds the 1° sample in $\chi^{(2)}$ due to the shift of the excitonic resonances. The black arrows mark the suppression of $\chi^{(2)}$ due to the quantum interference.

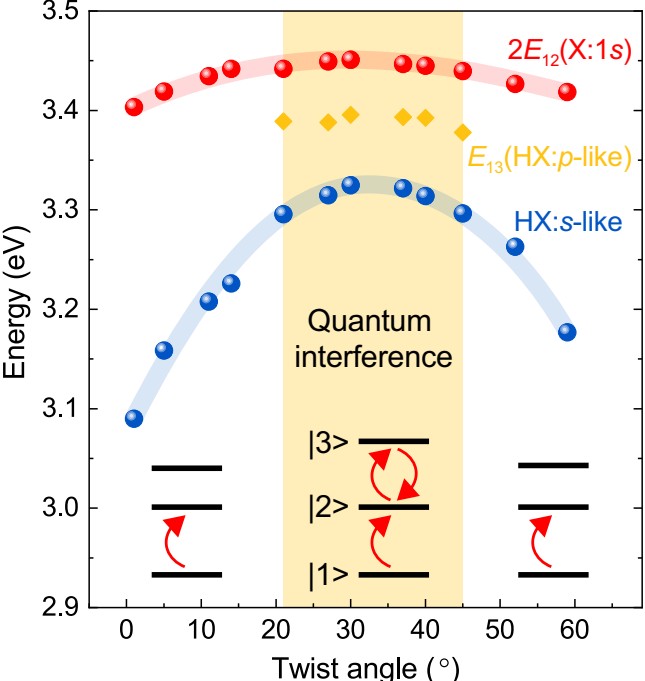

**Fig. 4 Control of quantum interference through twist angle in an excitonic three-level system in bilayer WSe₂ at 5 K.** Change of the A-exciton PL peak energy (red spheres), the HX PL peak energy (blue spheres), and the SHG quantum-interference dip energy (orange diamonds) with the twist angle. The lines serve as a guide to the eye. The effective excitonic three-level systems are illustrated for different twist angles. Since state |3⟩ shifts more strongly than state |2⟩, at angles approaching 0° or 60°, the $|1\rangle \rightarrow |2\rangle$ transition energy $E_{12}$ is much larger than the $|2\rangle \rightarrow |3\rangle$ transition energy $E_{23}$ so that quantum interference cannot occur within one excitation laser pulse (region not shaded in yellow).

quantum interference arises because of a significantly smaller value of $E_{12}$ than $E_{23}$ in monolayer MoSe₂. Assuming that the twist-angle dependence observed in bilayer WSe₂ also applies to bilayer MoSe₂, we expect the quantum interference to be recovered in twisted-bilayer MoSe₂. To test this idea, we fabricated bilayer MoSe₂ with a twist angle close to 0°, where the $E_{23}$ is expected to be tuned to its lowest value. Figure 5b shows the excitation-energy dependence of normalized SHG from this sample, which indeed reveals unambiguous signatures of quantum interference: the anti-crossing dependence of SHG emission energy on pump photon energy[17].

## Discussion

At present, hybridization of high-lying bands in twisted bilayers is challenging to analyze directly from DFT calculations because of the appearance of many folded bands in the reduced Brillouin zone. Twist-angle controlled quantum interference in SHG provides experimental access to the hybridization in specific high-lying bands. The HX emission has the advantage of being remarkably robust with regards to localized states, which generally appear below the bandgap. Moreover, intralayer excitons such as the A-exciton usually experience much shallower moiré potentials in comparison to interlayer excitons[33]. The HX should be an order of magnitude more sensitive to the moiré potential of the twisted bilayer compared with the A-exciton, and therefore offers further opportunities to explore moiré physics. Since the energy of the HX can be directly utilized as a metric of the strength of interlayer coupling, and the intra- and interlayer excitons are expected to have different spin-optical selection rules at different moiré sites[33], it would be interesting to explore the interplay between possible HX trapping by the moiré superlattice and its optical selection rules. We show that the optical nonlinearity in stacked bilayers is not only governed by the level of inversion symmetry but is also strongly modified by interlayer hybridization of electronic states. This non-trivial modulation of the frequency dependence of the second-order susceptibility with twist angle potentially offers a unique scheme for designing nonlinear materials. In contrast to quantum interference in

systems such as atomic gases[21,22], it is evidently straightforward to engineer the energies of ladder-type three-level systems in two-dimensional semiconductors. A perfectly degenerate three-level system can therefore be fabricated. The fact that one can detect SHG under continuous-wave conditions[20,34] in the limit of monolayers indicates that the light-matter interaction is intrinsically strong. In combination with the degeneracy, it may become possible to test quantum interference in the regime of low photon numbers to probe quantum nonlinearity—strong nonlinear interactions between individual photons[35]. Such quantum nonlinearity is considered promising for single-photon switches and transistors, and forms the basis of optical quantum logic[35]. The direct control of these systems through twist angle demonstrates the merit of 2D excitonic systems over well-established quantum systems based on atomic gases and color centers, and sets the basis for the development of quantum-excitonic nonlinear-optical devices.

## Methods

**Sample preparation**. WSe₂ monolayers were obtained through mechanical exfoliation from bulk crystals (HQ Graphene) onto commercial PDMS films (Gel-Pak, Gel-film® X4) using blue Nitto tape (Nitto Denko, SPV 224P)[23]. To fabricate twisted-bilayer WSe₂, we first stamped one part of a WSe₂ monolayer onto a silicon wafer with a thermal oxide layer of 285 nm thickness. After rotating the substrate orientation by the chosen stacking angle θ, the remaining part of the WSe₂ monolayer was then stamped on top. An optical microscope combined with translation stages was used to control the precise placement of the WSe₂ monolayer, while the substrate temperature was controlled through a small Peltier heating stage. In general, the substrate was heated to 65 °C before the layer on the PDMS stamp was attached to the layer already present on the wafer.

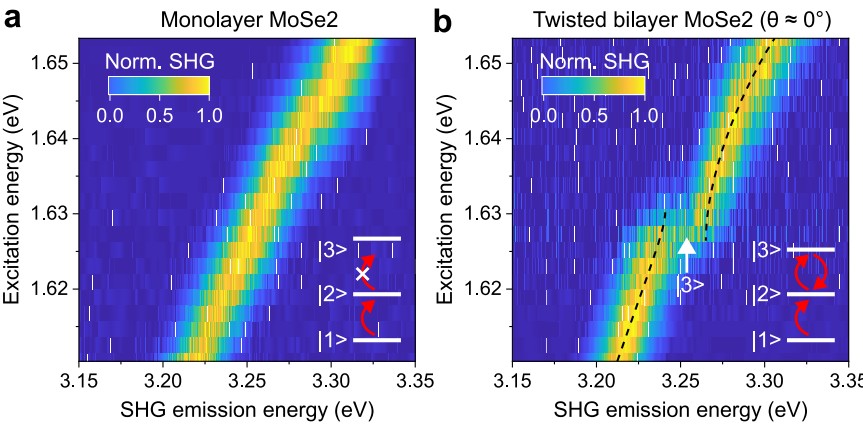

**Fig. 5 Recovering quantum interference in twisted-bilayer MoSe₂ at 5 K.** Dependence of normalized SHG spectra on excitation energy for monolayer MoSe₂ (**a**) and twisted-bilayer MoSe₂ with a twist angle close to 0° (**b**), with the proposed corresponding excitonic three-level systems illustrated as insets. The condition of an equidistant excitonic three-level system for single-pulse quantum interference is only fulfilled in twisted-bilayer MoSe₂ and not in monolayer MoSe₂.

**Optical spectroscopy**. The PL of twisted-bilayer WSe₂ was measured using the 488 nm line of an argon-ion laser (Spectra-Physics, 2045E). UPL of twisted-bilayer WSe₂ was measured using the narrow-band 720 nm emission of a tunable continuous-wave laser (Sirah, Matisse CR). SHG was measured by excitation using a tunable Ti:sapphire laser with 80 fs pulse length (Newport Spectra-Physics, Mai Tai XF, 80 MHz repetition rate). After passing through a 488 nm long-pass edge filter (Semrock, LP02-488RU) or a 680 nm short-pass filter (Semrock, FF01-680SP), the PL, UPL, or SHG signal was dispersed by a grating of 600 grooves/mm or 1200 grooves/mm, and detected by a cooled CCD camera (Princeton Instruments, PIXIS 100). The lasers were focused through a 0.6 numerical aperture microscope objective (Olympus, LUCPLFLN ×40) and a 1.3-mm-thick fused silica window onto the samples mounted under high-vacuum conditions on the cold finger of a liquid-helium cryostat (Janis, ST-500). A 50:50 non-polarizing plate beam splitter (Thorlabs, BSW26R) was used to separate the incident optical path and the signal detection path. To obtain an acceptable signal-to-noise ratio of the A-exciton PL from twisted bilayers in Fig. 1d, the power of the laser at 488 nm was set to 50 μW and the integration time was chosen as 10 s. To measure the high-lying exciton UPL in Fig. 1d, the laser was tuned to 720 nm with a power of 50 μW, and the integration time was set to 100 s. The 600 grooves/mm grating was used for UPL and PL measurements. To measure the excitation-wavelength dependent SHG of twisted-bilayer WSe₂ in Figs. 2d and 3, we scanned the excitation wavelength from 715 nm to 750 nm with 1 nm steps while keeping a constant power of 1 mW. A 1200 grooves/mm grating was used and the integration time was typically set to 10 s. Since the second-order susceptibility drops significantly from 0° to 60° due to the recovery of inversion symmetry, the integration time was increased to 100 s for the measurement of 59° twisted-bilayer WSe₂. To obtain the excitation-wavelength-dependent SHG of monolayer and twisted-bilayer MoSe₂ in Fig. 5, we scanned the excitation wavelength from 750 to 770 nm in 1 nm steps while maintaining a constant power of 0.25 mW. The integration time was set to 100 s and 1200 grooves/mm grating was used. The powers were measured between the objective and the beam splitter.

**Estimation of effective second-order susceptibility**. The second-order susceptibility of the twisted-bilayer WSe₂ is measured against a standard z-cut α-quartz crystal following ref. [36]. We determine the effective sheet second-order susceptibility $\left|\chi_{\text{sheet}}^{(2)}\right|$ of twisted bilayers from $\frac{\left|\chi_{\text{sheet}}^{(2)}\right|}{\left|\chi_{\text{quartz}}^{(2)}\right|} = \frac{c}{4\omega[n(\omega)+n(2\omega)]}\sqrt{\frac{I_{\text{sample}}}{I_{\text{quartz}}}}$, where $\chi_{\text{quartz}}^{(2)} = 0.6$ pm/V, $c$ is the speed of light, $\omega$ is the excitation (i.e., fundamental) frequency, $n$ is the refractive index of the quartz, $I_{\text{sample}}$ is the SHG intensity (the integral of the SHG spectrum) of the sample, and $I_{\text{quartz}}$ is the SHG intensity measured from the quartz crystal. Finally, the effective-volume second-order susceptibility can be obtained by $\left|\chi_{\text{volume}}^{(2)}\right| = \left|\chi_{\text{sheet}}^{(2)}\right|/t_{\text{sample}}$, where $t_{\text{sample}}$ is the thickness of the sample, taken to be 1.4 nm for all bilayers.

## Data availability
The raw data that support the plots within this paper and the other findings of this study are available from the corresponding authors upon reasonable request.

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

## Acknowledgements

We thank J. Kunstmann and S. Brem for helpful discussions, and S. Krug for technical assistance. Financial support is gratefully acknowledged from the Deutsche For-schungsgemeinschaft (DFG, German Research Foundation) SPP 2244 (Project-ID 443378379 and 443416183), and SFB 1277 (Project-ID 314695032) projects B03 and B05. P.E.F.J. was supported by a Capes-Humboldt Research Fellowship of the Alexander von Humboldt Foundation and Capes (Grant No. 99999.000420/2016-06). B.P. and B.M. were supported by the Gianna Angelopoulos Programme for Science, Technology, and Innovation and the Winton Programme for the Physics of Sustainability. M.G. acknowledges support by VEGA 1/0105/20 and VVGS-2019-1227.

## Author contributions

K.-Q.L. conceived and performed the experiments; J.M.B. and K.L. fabricated the samples; P.E.F. J., B.P., B.M., M.G., and J.F. performed the DFT calculations; K.L., S.B., and J.M.L. analyzed the data and wrote the paper, with input from all authors.

## Funding

## Competing interests

The authors declare no competing interests.
