## [Peer Review File · Nature Communications]

Reviewers' Comments:

Reviewer #1:

Remarks to the Author:

As stated in my first report, I think this is a very interesting paper that looks to the effect of the twist angle in the quantum interference of excitons monitored by SHG. The work extends the work done by the authors in previous publications providing insightful new results. The new version has been improved a lot and the authors have satisfactorily answered all the questions and considered the suggestions I made in my previous report (and most of the concerns raised by the other referees). They have provided compelling evidence for the control of EIT through the twisting angle, and even if the theory analysis could be enhanced and made more quantitative, the level of discussion they provide is enough to support the experimental findings. Therefore, I think the present version of the work is suitable for publication in *Nature Communications*.

Reviewer #2:

Remarks to the Author:

The paper titled "Twist-angle engineering of excitonic quantum interference and optical nonlinearities in stacked 2D semiconductors" shows an experimental investigation of the influence of a twist angle between monolayers of WSe₂ on its excitons and the corresponding SHG emission. The authors reveal that high-lying excitons are an order of magnitude more sensitive to the moiré potential compared with band-edge excitons, which enables the control of χ^2 nonlinearity and the associated quantum interference in twistrionics.

The paper is very well organised, presents novel results, is well written and contains, to the best of my knowledge, a good list of appropriate references. The figures are clear and easy to understand. The authors have also performed a major revision, addressing all concerns of the referees when transferring to *Nature Communications*.

One important shortcoming of this work is the way the impact is discussed, given a broad audience of *Nature Communications*. Although the novelty of the presented results is explained very well, the importance may be difficult to understand for readers beyond the highly specialised area of nonlinear twistrionics. Why is it important to modulate the frequency dependence of the second-order susceptibility? What would be interesting to explore in moiré physics using this approach? Why would one benefit from testing quantum interference in the regime of low photon numbers in this case? Are there any potential practical applications of this discovery?

Reviewer #3:

Remarks to the Author:

This work reports the optical observation of the high-lying exciton (HX) in twisted-bilayer WSe₂. The authors use mainly SHG as a probe of this exciton state and the so-called quantum interference are investigated by changing the excitation photon energy.

I agree with the previous reviewers that the novelty of this work is the excitonic states at around 3.2 eV and their sensitivity to the twist angle. From this point of view, quantum interference could be one of the examples to show the significance of these new states. The current form of the

manuscript somehow emphasizes too much of the interference side. For me, it is more reasonable if the title reads like "twist-angle engineering of high-lying exciton and optical nonlinearities ..."

Similar to the well-established rich excitonic fine structures (neutral, charged excitons, dark exciton and their charged versions, spin-singlet and triplet charged excitons, etc.) in hBN encapsulated WSe₂, any well-addressed interlayer states in stacked 2D materials provides more channels to understand and engineer the vdW structures. The topic and findings of this work are therefore suitable for the audiences of Nature Communications in the field of 2D materials. I am positive with the publication given the authors could provide extra basic measurements to address the following concerns.

1. There are a couple of simple, yet crucial questions need to be answered for the evidence of HX in TMD twisted bilayers: What is the power dependence of the UPL of HX? Does the HX generally exist in MoSe₂ bilayers or other heterobilayers? What are the polarization properties of these peaks? Is the HX observable under normal PL conditions (excitation photon energy larger than 3.4 eV)? What is the estimated binding energy and size of the s state?
2. When the SHG energy is close to the HX, the excitonic resonance could also play a role apart from the interference. What is the interplay between the two mechanisms? Does the s-like HX modified SHG profile appear?
3. The p-like HX state is still abrupt and vague for me. Is this means the higher n=2 p state? Please specify and give more discussions.
4. In the supplements there are no figures "Fig. S1a", "Fig. S1b" and "Fig. S3" is not the "calculated distances" as mentioned in Note 1.

Review Comments (in black) and Author Responses (in dark blue)

Reviewer #1 (Remarks to the Author):

Reviewer #1: “As stated in my first report, I think this is a very I think interesting paper that looks to the effect of the twist angle in the quantum interference of excitons monitor by SHG. The work extends the work done by the authors in previous publications providing insightful new results. The new version has been improved a lot and the authors has satisfactory answered all the questions and considered the suggestions I made in my previous report (and most of the concerns raised by the other referees). They have provided compelling evidence for the control of EIT through the twisting angle, and even if the theory analysis could be enhanced and make more quantitative, the level of discussion they provide is enough to support the experimental findings. Therefore, I think the present version of the work is suitable for publication In Nature Communications.”

Response 1: We thank the reviewer for the appreciation of the work and the support for publication.

Reviewer #2: “The paper titled “Twist-angle engineering of excitonic quantum interference and optical nonlinearities in stacked 2D semiconductors” shows an experimental investigation of the influence of a twist angle between to monolayers of WSe2 on its excitons and the corresponding SHG emission. The authors reveal that high-lying excitons are an order of magnitude more sensitive to the moiré potential compared with band-edge excitons, which enables the control of χ^2 nonlinearity and the associated quantum interference in twistrionics.

The paper is very well organised, presents novel results, is well written and contains, to the best of my knowledge, a good list of appropriate references. The figures are clear and easy to understand. The authors have also performed a major revision, addressing all concerns of the referees when transferring to Nature Communications.”

Response 2.1: We are grateful to the reviewer for the appreciation of the work and the support for publication.

One important shortcoming of this work is the way the impact is discussed, given a broad audience of Nature Communications. Although the novelty of the presented results is explained very well, the importance may be difficult to understand for readers beyond the highly specialised area of nonlinear twistrionics. Why is it important to modulate the frequency dependence of the second-order susceptibility? What would be interesting to explore in moiré physics using this approach? Why would one benefit from testing quantum interference in the regime of low photon numbers in this case? Are there any potential practical applications of this discovery?

Response 2.2: We appreciate the series of questions the reviewer offered to help us to discuss the broadened impact of our work. We have now added the following points to the discussions: “Resonant enhancement of the second-order susceptibility by excitonic transitions is usually limited to a specific range of frequencies. The ability to tune resonances allows the frequency dependence of the second-order susceptibility to be chosen to match a particular desired frequency range.”; “Moreover, intralayer excitons such as the A-exciton usually experience much shallower moiré potentials in comparison to interlayer excitons. The HX should be an order of magnitude more sensitive to the moiré potential of the twisted bilayer compared with the A-exciton, and therefore offers new opportunities to explore moiré physics. Since the energy of the HX can be directly utilized as a metric of the strength of interlayer coupling, and the intra- and interlayer excitons are expected to have different spin-optical selection rules at different moiré sites, it would be interesting to explore the interplay between possible HX trapping by the moiré superlattice and its optical selection rules.”; “A perfectly degenerate three-level system can therefore be fabricated. The fact that one can detect SHG

under continuous-wave conditions in the limit of monolayers indicates that the light-matter interaction is intrinsically extremely strong. In combination with the degeneracy, it may become possible to test quantum interference in the regime of low photon numbers to probe quantum nonlinearity – strong nonlinear interactions between individual photons. Such quantum nonlinearity is considered promising for single-photon switches and transistors, and forms the basis of optical quantum logic.”. Regarding the potential applications of this discovery, we already discussed in the manuscript that our finding offers a new method to assess the angle of twisting, complementary to the conventional technique based on polarization-resolved optical SHG. A longer-term realistic vision lies in building optoelectronic devices that exploit the quantum optical phenomenon. Since atomic-gas systems cannot be directly combined with optoelectronic devices, the excitonic system in TMDC van der Waals heterostructures offers new opportunities.

Reviewer #3 (Remarks to the Author):

Reviewer #3: “This work reports the optical observation of the high-lying exciton (HX) in twisted-bilayer WSe₂. The authors use mainly SHG as a probe of this exciton state and the so-called quantum interference are investigated by changing the excitation photon energy.

I agree with the previous reviewers that the novelty of this work is the excitonic states at around 3.2 eV and their sensitivity to the twist angle. From this point of view, quantum interference could be one of the examples to show the significance of these new states. The current form of the manuscript somehow emphasizes too much of the interference side. For me, it is more reasonable if the title reads like “twist-angle engineering of high-lying exciton and optical nonlinearities ...

Similar to the well-established rich excitonic fine structures (neutral, charged excitons, dark exciton and their charged versions, spin-singlet and triplet charged excitons, etc.) in hBN encapsulated WSe₂, any well-addressed interlayer states in stacked 2D materials provides more channels to understand and engineer the vdW structures. The topic and findings of this work are therefore suitable for the audiences of Nature Communications in the field of 2D materials. I am positive with the publication given the authors could provide extra basic measurements to address the following concerns.”

Response 3.1: We thank the reviewer for this appreciation of our work and the support regarding its publication. Although we agree with the reviewer that the high twist-angle sensitivity of the HX is a central novel point of the paper, we have to emphasize that the control of the quantum interference is equally novel since this degree of freedom does not exist in atomic-gas or crystal defect-based systems where the energy of the state is fixed. With this work, we intend to demonstrate a clear benefit of the excitonic multilevel system for exploring quantum-optical phenomena over conventional atomic-gas-based systems. For this reason, we would prefer to keep the present title as it is, to highlight the remarkable tunability of quantum interference and also to attract the attention of readers from the quantum optics research community. We address the concerns below.

1. There are a couple of simple, yet crucial questions need to be answered for the evidence of HX in TMD twisted bilayers: What is the power dependence of the UPL of HX? Does the HX generally exist in MoSe₂ bilayers or other heterobilayers? What are the polarization properties of these peaks? Is the HX observable under normal PL conditions (excitation photon energy larger than 3.4 eV)? What is the estimated binding energy and size of the s state?

Response 3.2: Indeed, these are crucial questions that we investigated in detail in Ref. 16 with monolayer WSe₂ from both a theoretical and an experimental perspective. The power dependence

of the HX UPL is not quadratic due to the bleaching of the A exciton transition, which constrains the upconversion yield based on the Auger-like exciton-exciton annihilation mechanism. The exact power-law exponent depends on the range of pump powers (i.e. the range of exciton densities), in analogy to the discussion in Ref. 22 and also in Ref. 16. Yes, we confirm that the HX also exists in MoSe₂ bilayers, but we would prefer to discuss this in detail elsewhere since there are several non-trivial complications to the observation. Whether the HX exists in other heterobilayers remains to be investigated – these are not entirely trivial experiments because the HX constitutes a new excitonic state around the K points. To selectively probe it through UPL, the twist angle and band alignment of heterobilayers requires careful design to ensure the energy and momentum conservation necessary for the Auger-like exciton-exciton annihilation process. As demonstrated in the Ref. 16, the polarization of the HX luminescence is in-plane, similar to the case of the A exciton. We provide additional one-photon PL measurements (i.e. recorded under conventional “Stokes-type” PL conditions) of the HX below in Figure 1R. Here, the bilayer WSe₂ is excited at a wavelength of 325 nm (~3.82 eV) by a continuous-wave Helium-Cadmium laser. The HX is indeed observable but its overall structure tends to be obscured under normal PL conditions due to the broad background, which inevitably arises both from the substrate and from WSe₂ transitions away from the K points in momentum space. In contrast, pumping the A exciton transition selectively excites the HX in the vicinity of the K points and thus gives rise to a background-free HX PL spectrum which has more structure. The binding energy of the HX in monolayer WSe₂ was estimated to be ~0.6 eV from the calculations in Ref. [16], and the root mean square radius of the exciton was calculated to be 1.2 nm, i.e. smaller than the 1.5 nm concluded for the A exciton. Nevertheless, we expect that these numbers will change in the twisted bilayers due to the change in band curvature and the dielectric screening. Unfortunately, theoretical treatment of such systems is still exceedingly challenging. We have added additional discussions to the manuscript and included Figure R1 in the Supplementary Information to address these queries.

Figure 1R. One-photon PL and UPL of the HX from hBN encapsulated bilayer WSe₂ on a sapphire substrate. The one-photon PL of the HX was measured by exciting the bilayer WSe₂ with a 325 nm (3.81 eV) continuous-wave Helium-Cadmium laser. An aluminum reflective objective (36 \times , NA = 0.5, Beck Optronic Solutions) was used to focus the laser and collect the PL signals, and a D-shape UV-enhanced aluminum mirror was used in place of the beam splitter.

2. When the SHG energy is close to the HX, the excitonic resonance could also play a role apart from the interference. What is the interplay between the two mechanisms? Does the *s*-like HX modified SHG profile appear?

Response 3.3: Indeed, the excitonic resonance can enhance the SHG, provided that quantum interference does not occur. However, quantum interference suppresses the SHG. We observe a broad enhancement of the SHG efficiency (Fig. 3) around the energy where the *s*-like HX is found, i.e. where the radiative transition of the HX appears. No change of the SHG spectral profile was observed at these frequencies.

3. The *p*-like HX state is still abrupt and vague for me. Is this means the higher $n=2$ *p* state? Please specify and give more discussions.

Response 3.4: The *p*-like HX state is also discussed in detail in Ref. 16. With this terminology we imply an HX state that has a different parity from the emissive "*s*-like" HX. The *p*-like state is identified clearly by two-photon absorption spectroscopy in Ref. 16. We now explicitly state this fact in the present manuscript to aid understanding here without the reader having to delve into Ref. 16. However, we do not have sufficient evidence as yet to assign this "*p*-like" two-photon-active state conclusively to the $2p$ state of the HX; conclusive assignment will necessitate very challenging magnetoPL spectroscopy in the UV spectral range. We have added the necessary discussions to the manuscript so that it will be much clearer to readers now.

4. In the supplements there are no figures "Fig. S1a", "Fig. S1b" and "Fig. S3" is not the "calculated distances" as mentioned in Note 1.

Response 3.5: We apologize for the inconsistencies that arose during the last revision. It was "Fig. 2a" instead of "Fig. S1a" and "Fig. S1b", and it should be "Supplementary Fig. 4" instead of "Fig. S3", where we showed the calculated interlayer distance. We have gone through the figures and text again to ensure consistency.

Reviewers' Comments:

Reviewer #2:

Remarks to the Author:

The manuscript appears to have been improved substantially, addressing all major concerns. I can recommend this submission for publication.

Reviewer #3:

Remarks to the Author:

I have carefully read the answers and the updated manuscript. The authors have addressed my concerns with clear discussions and data. This work is of good novelty and will bring the UPL, HX and control of the interference to the audiences' attention. I think this work is now ready for publication.